# Maternal pre-pregnancy body mass index and related factors: A cross-sectional analysis from the Japan Environment and Children's Study

**Yasuaki Saijo**[1]*, **Eiji Yoshioka**[1], **Yukihiro Sato**[1], **Yuki Kunori**[1], **Tomoko Kanaya**[1], **Kentaro Nakanishi**[2], **Yasuhito Kato**[2], **Ken Nagaya**[3], **Satoru Takahashi**[4], **Yoshiya Ito**[5], **Hiroyoshi Iwata**[6], **Takeshi Yamaguchi**[6], **Chihiro Miyashita**[6], **Sachiko Itoh**[6], **Reiko Kishi**[6], **the Japan Environment and Children's Study (JECS) Group**[¶]

1 Department of Social Medicine, Asahikawa Medical University, Sapporo, Hokkaido, Japan, 2 Department of Obstetrics and Gynecology, Asahikawa Medical University, Asahikawa, Hokkaido, Japan, 3 Division of Neonatology, Perinatal Medical Center, Asahikawa Medical University Hospital, Asahikawa, Hokkaido, Japan, 4 Department of Pediatrics, Asahikawa Medical University, Asahikawa, Hokkaido, Japan, 5 Faculty of Nursing, Japanese Red Cross Hokkaido College of Nursing, Kitami, Hokkaido, Japan, 6 Center for Environmental and Health Sciences, Hokkaido University Sapporo, Hokkaido, Japan

¶ Membership of the JECS group is provided in the Acknowledgments.
* y-saijo@asahikawa-med.a.c.jp

**Data Availability Statement:** Data are unsuitable for public deposition due to ethical restrictions and legal framework of Japan. It is prohibited by the Act

## Abstract

Socioeconomic status and smoking are reportedly associated with underweight and obesity; however, their associations among pregnant women are unknown. This study aimed to investigate whether socioeconomic factors, namely educational attainment, household income, marital status, and employment status, were associated with pre-pregnancy body mass index (BMI) categories, including severe-moderate underweight (BMI $\leq$ 16.9 kg/m$^2$), mild underweight (BMI, 17.0–18.4 kg/m$^2$), overweight (BMI, 25.0–29.9 kg/m$^2$), and obese (BMI $\geq$ 30.0 kg/m$^2$) among Japanese pregnant women using data from the Japan Environment and Children's Study (JECS). In total, pregnant women were included 96,751. Age- and parity-adjusted multivariable multinomial logistic regression analyses assessed socioeconomic factors and smoking associations with falling within abnormal BMI categories (normal BMI as the reference group). Lower education and lower household were associated with overweight and obesity, and, especially, lowest education and household income had relatively higher point estimate relative ratios (RRs) of 3.97 and 2.84, respectively. Regarding the risks for underweight, however, only junior high school education had a significantly higher RR for severely to moderately underweight. Regarding occupational status, homemakers or the unemployed had a higher RR for severe-moderate underweight, overweight, and obesity. Unmarried, divorced, or bereaved women had significantly higher RRs for mildly underweight status. Quitting smoking early in pregnancy/still smoking had higher RRs for all four not having normal BMI outcomes; however, quitting smoking before pregnancy had a higher RR only for obese individuals. Lower educational attainment and smoking are essential intervention targets for obesity

on the Protection of Personal Information (Act No. 57 of 30 May 2003, amendment on 9 September 2015) to publicly deposit the data containing personal information. Ethical Guidelines for Medical and Health Research Involving Human Subjects enforced by the Japan Ministry of Education, Culture, Sports, Science and Technology and the Ministry of Health, Labour and Welfare also restricts the open sharing of the epidemiologic data. All inquiries about access to data should be sent to: jecs-en@nies.go.jp. The person responsible for handling enquiries sent to this e-mail address is Dr Shoji F. Nakayama, JECS Programme Office, National Institute for Environmental Studies.

**Funding:** This study was funded by the Ministry of Environment, Japan. The findings and conclusions of this article are solely the responsibility of the authors and do not represent the government's official views. The funders had no role in study design, data collection and analysis, decision to publish, or preparation of the manuscript.

**Competing interests:** The authors have declared that no competing interests exist.

and severe-moderate underweight prevention in younger women. Lower household income is also a necessary target for obesity.

## Introduction

Underweight among young women has been a significant health issue in Japan for several decades [1, 2], and approximately 20% of women aged 20–39 years have a low body mass index (BMI: <18.5 kg/m$^2$) [3]. Underweight younger women have a higher risk of low bone density, and underweight middle-aged and elderly populations have a higher risk of bone fractures and mortality [4–6]. Furthermore, being underweight among pregnant women affects obstetric outcomes, including preterm birth and small for gestational age [7].

Obesity among young Japanese women is not as prevalent as in other countries; however, the obesity rate among Japanese women has increased with age, and obesity is a well-known risk factor for morbidity, mortality, and lower quality of life [3, 8, 9]. Similar to underweight, obesity in pregnant women is a risk factor for obstetric outcomes, including elective cesarean delivery, emergency cesarean delivery, gestational diabetes, gestational hypertension, and an increased risk of admission to the newborn intensive care unit [7].

Socioeconomic status (SES), including education, income, marital status, and employment, is associated with being underweight and obesity [10–16]. However, regarding pregnant women, an age and parity adjustment analysis among a relatively large population has only been reported among French pregnant women (N = 25,566) [17], and the study reported that education and income were inversely associated with overweight and obesity. Income, but not education, was inversely and mildly associated with being underweight.

Smoking is prevalent among people with lower SES in most developed countries [18]; a study among men in England aged 31–65 reported that former smokers had a higher risk of obesity but that current smokers had a lower risk. These significances disappeared in the subgroup analyses among those aged 40 or under [19]. A study on Cambodian women aged 15–49 found no significant relationship between smoking and obesity [12]. A study among Indian women aged 18–49 reported that current smoking had a protective relationship with obesity and an aggravating relationship with underweight [20]. Meanwhile, a study among Indonesian men and women aged 18–103 reported that current smoking had a protective relationship with obesity but no relationship with being underweight [21]. A study among Koreans aged 25–69 years reported that current smoking was related to underweight among men, and ex-smokers were more likely to be underweight than non-smokers among women [22]. Thus, the association between smoking, underweight, and obesity seems to vary across sex, age, and country. However, these associations have not been investigated in pregnant women using multivariate-adjusted analyses.

The effect of smoking on BMI has been inconsistent in previous studies. Although associations between SES and smoking have been reported in underweight and obese individuals, they may vary depending on sex, age, and country [23]. Furthermore, underweight and obese women have severe grades and severe underweight and obesity are related to more adverse obstetric outcomes in pregnant women [24, 25] and higher mortality in the general population [26, 27]. However, associations between SES and smoking simultaneously in underweight and obese pregnant women with multivariable adjustments have not been reported. Obesity and being underweight during pregnancy increase the risk of adverse obstetric outcomes. Identifying related factors can help prevent these outcomes. This study aimed to investigate whether SES like educational attainment, household income, marital status, and employment status were

associated with pre-pregnancy body mass index (BMI) categories, including severe-moderate underweight (BMI $\leq$ 16.9 kg/m$^2$), mild underweight (BMI, 17.0–18.4 kg/m$^2$), overweight (BMI, 25.0–29.9 kg/m2), and obese (BMI $\geq$ 30.0 kg/m$^2$) among Japanese pregnant women using a large scale cohort data from the Japan Environment and Children's Study (JECS) [28, 29].

## Materials and methods

### Participants

The JECS is an ongoing nationwide, large-scale prospective birth cohort study in Japan, aiming to identify environmental factors affecting children's health and development [28, 29]. The recruitment of pregnant women and follow-up of their children were carried out in 15 Regional Centers located in all geographical areas of Japan (Hokkaido, Miyagi, Fukushima, Chiba, Kanagawa, Koshin, Toyama, Aichi, Kyoto, Osaka, Hyogo, Tottori, Kochi, Fukuoka, and South Kyushu/Okinawa). The protocol for analyzing the data in this study was prepared after the relevant data was collected.

Pregnant women were recruited during the early stages of pregnancy, and 103,057 pregnancies were registered between January 2011 and March 2014. Written informed consent was obtained from all participants prior to data collection. The inclusion criteria were as follows: 1) Pregnant women whose expected delivery dates were between August 2011 and March 2014, 2) pregnant women who resided in one of the study areas selected by the Regional Centres at the time of recruitment, and who were expected to reside continually in Japan for the foreseeable future, and 3) pregnant women who visited a cooperating health care provider selected by the Regional Centre or local government offices to obtain a Mother-Child Health Handbook in a study area during the recruiting period. The exclusion criteria were as follows: 1) Pregnant women who did not consent to participate in the study, 2) pregnant women who showed difficulty in comprehending the study procedures or filling out the questionnaires without support, and 3) pregnant women who were reportedly not accessible at the time of delivery (e.g., women who planned to give birth outside the study area). Recruitment activities were conducted at healthcare providers and local government facilities to identify eligible women in the study areas. However, the recruitment was not entirely random. Our team made every effort to reach out to as many eligible women in the study areas as possible. The child coverage was estimated to be approximately 45% of the studied areas [28, 29]. After excluding pregnancies involving the same woman, the study included 97,410 unique pregnancies. After excluding participants with missing BMI data, the final number of participants was 96,751 (Fig 1). We used the jecs-ta-20190930 dataset from the JECS, which was the JECS registry, the University Hospital Medical Information Network (UMIN) 000030786 (UMIN Clinical Trials Registry).

### Ethics statement

The JECS protocol was reviewed and approved by the Institutional Review Board on Epidemiological Studies of the Ministry of the Environment and the Ethics Committees of all participating institutions. The JECS was conducted by the Declaration of Helsinki and other national regulations. Written informed consent was obtained from all participants.

### Outcomes

Maternal height and pre-pregnancy weight were obtained from medical records. If missing, these data were obtained from self-reports. Pre-pregnancy BMI was calculated as maternal pre-pregnancy weight (kg) divided by the square of maternal height (m$^2$) obtained from medical record transcriptions or self-reports. The participants were categorized based on their pre-

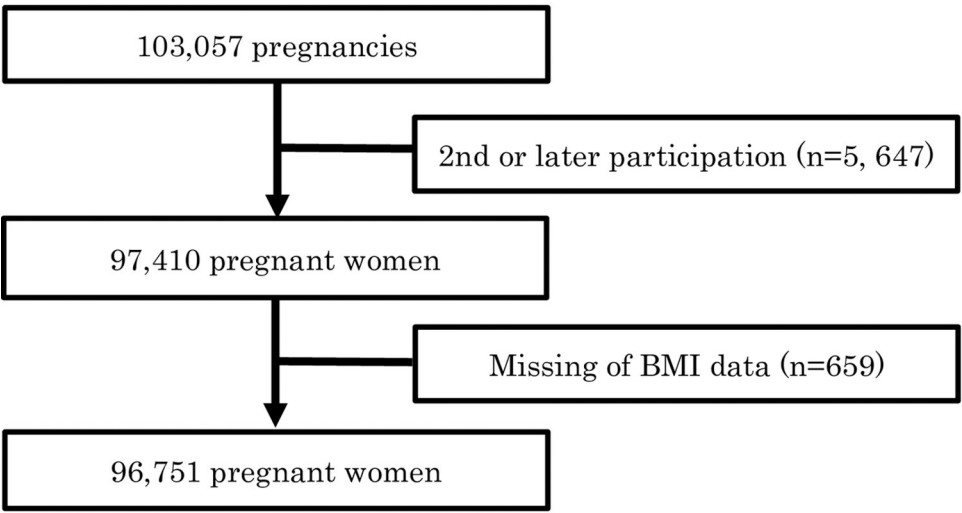

**Fig 1. Flowchart of the study.**

pregnancy BMI as follows: severe-moderate underweight (BMI ≤ 16.9 kg/m$^2$), mild underweight (BMI, 17.0–18.4 kg/m$^2$), normal weight (BMI, 18.5–24.9 kg/m$^2$), overweight (BMI, 25.0–29.9 kg/m$^2$), and obese (BMI > 30.0 kg/m$^2$) [24, 30, 31]. The normal pre-pregnancy BMI group was used as the reference group for statistical analyses.

### Socioeconomic status, smoking status, and covariates

This study determined SES by mothers' educational attainment, household income, occupational status, and marital status. During pregnancy, questionnaires were distributed to the enrolled mothers during the first trimester (T1; if participation was delayed, they were distributed during the second or third trimester) and the second or third trimester (T2). The latter included questions about mothers' educational attainment, categorized as ≤ 9 years (EDC1: junior high school), 10 ≤12 years (EDC2: high school), 13–15 years (EDC3: technical junior college, technical/vocational college, or associate degree), or ≥16 years (EDC4: bachelor's or postgraduate degree). The T2 questionnaire also included questions on annual household income, categorized as ≤199, 200–399, 400–599, 600–799, 800–999, and ≥1000 million yen.

The T1 questionnaire for mothers included questions on marital, occupational, and smoking statuses. Marital status was classified as married, unmarried, divorced, or bereaved. Occupational status was classified as employed, homemaker, unemployed, or student. Smoking status was categorized as never smoking, quitting before pregnancy, or quitting smoking during early pregnancy/still smoking.

Maternal age and parity were selected as covariates (confounders) based on previous research on pregnant women [17, 32], and data were transcribed from the medical records. Maternal age was categorized as ≤19, 20–24, 25–29, 30–34, 35–39, or ≥40 years, and parity was ranked as zero, one, two, or more.

### Statistical analysis

Because there were some missing values, except for BMI, to impute missing values for participants with missing data (4.1% of data were missing), the information was replaced using multiple imputations by a chained equation (25 imputed datasets) based on the assumption that

data were missing at random. The variables included in the imputation model mentioned above are all variables.

We then constructed an adjusted multinomial logistic regression model to assess the associations between socioeconomic factors, smoking, and not falling within normal BMI categories (with normal BMI as the reference group). The models included educational attainment, household income, marital status, occupational status, smoking status, and covariates (age and parity). We also performed an adjusted multinomial regression analysis with the same model, using a complete dataset as a sensitivity analysis.

Two-sided P-values less than 0.05 were considered statistically significant. All analyses were performed using Stata statistical software (version 17.0) for Windows (StataCorp, College Station, TX, USA).

## Results

The percentages of severely moderate underweight, mildly underweight, normal weight, overweight, and obese participants were 2.7%, 13.0%, 73.4%, 8.3%, and 2.6%, respectively (Table 1).

Table 2 presents the results of the multivariate multinomial logistic regression analysis of the explanatory variables for not having a normal BMI. Younger pregnant women had significantly higher relative ratios (RRs) for severe-moderate and mild underweight and significantly lower RRs for overweight and obese women. Pregnant women with parity two or more had significantly lower RRs for severe-moderate and mild underweight and a significantly higher RR for overweight. Quitting smoking early pregnancy/still smoking had significantly higher RRs for all not having a normal BMI, and quitting smoking before pregnancy had significantly lower RRs for severe-moderate and mildly underweight individuals and a significantly higher RR for obese individuals. Regarding marital status, unmarried, divorced, or bereaved individuals had a significantly higher RR for mildly underweight status. Regarding occupational status, homemakers or unemployed individuals had a significantly higher RR for severe-moderate, mildly underweight, and obese individuals. Lower income (≤199, 200–399, 400–599 million yen) had significantly higher RRs for overweight and obese, but the 200–399, 400–599, and 600–899-million-yen groups had significantly lower RRs for mild underweight, and 400–599 had a significantly lower RR for severe-moderate underweight. Compared to ECD4, ECD1, 2, and 3 had significantly higher RRs for overweight and obesity, ECD2 and 3 had significantly lower RRs for mild underweight, and ECD1 had a significantly lower RR for severe-to-moderate underweight. The results were nearly similar in the sensitivity analysis of the complete dataset (see S1 Table). However, three positive RRs changed to negative (≤19 years old for underweight, ≥ 40 years for obese, and 400–599 million yen for mildly underweight), and one negative RR changed to positive (divorced or bereaved for obese).

## Discussion

In this study, lower education and lower household income were associated with overweight and obesity, and, especially, lowest education and household income had relatively higher point estimate RRs of 3.97 and 2.84, respectively. Regarding the risks for being underweight, however, only ECD1 was associated with a higher risk of severe to moderate underweight. Regarding occupational status, homemakers or unemployed was associated with a higher risk of severe-moderate underweight, overweight, and obesity. Regarding marital status, only mildly underweight individuals had a significant risk of being unmarried, divorced, or bereaved. Quitting smoking early in pregnancy/still smoking was a risk factor for all four not having a normal BMI, but quitting smoking before pregnancy was the only risk factor for

**Table 1. Characteristics of the participants by body mass index class (N = 96,751).**

| | N | Severe-moderate underweight (BMI≤16.9 kg/m²) (N = 2,640, 2.7%) | Mild underweight (BMI: 17.0–18.4 kg/m²) (N = 12,564, 13.0%) | Normal (BMI: 18.5–24.9 kg/m²) (N = 70,995, 73.4%) | Overweight (BMI: 25.0–29.9 kg/m²) (N = 8,057, 8.3%) | Obesity (BMI≥30.0 kg/m²) (N = 2,495, 2.6%) |
|---|---|---|---|---|---|---|
| | | % | % | % | % | % |
| Age (years) | | | | | | |
| ≤19 | 1,148 | 6.1 | 17.1 | 69.3 | 6.5 | 1.0 |
| 20–24 | 9,877 | 4.3 | 16.5 | 69.7 | 7.5 | 2.0 |
| 25–29 | 27,731 | 3.2 | 14.3 | 72.8 | 7.4 | 2.3 |
| 30–34 | 32,889 | 2.5 | 12.5 | 74.1 | 8.3 | 2.6 |
| 35–39 | 19,731 | 1.9 | 10.6 | 75.1 | 9.5 | 3.0 |
| ≥40 | 3,429 | 1.3 | 9.2 | 74.6 | 11.6 | 3.4 |
| Missing | 1,946 | 2.4 | 12.0 | 70.8 | 10.0 | 4.8 |
| Parity | | | | | | |
| 0 | 40,199 | 3.1 | 13.9 | 73.7 | 7.2 | 2.2 |
| 1 | 35,234 | 2.5 | 12.8 | 73.2 | 8.8 | 2.6 |
| ≥2 | 18,843 | 2.3 | 11.3 | 73.0 | 10.0 | 3.4 |
| Missing | 2,475 | 3.2 | 15.0 | 73.6 | 6.6 | 1.7 |
| Smoking | | | | | | |
| Never-smoking | 54,459 | 2.6 | 13.2 | 74.6 | 7.6 | 2.0 |
| Quitting smoking before pregnancy | 21,836 | 2.2 | 11.7 | 74.3 | 8.9 | 3.0 |
| Quitting smoking early pregnancy/ still smoking | 17,204 | 3.8 | 13.8 | 68.8 | 10.1 | 3.6 |
| Missing | 3,252 | 3.6 | 13.9 | 70.9 | 8.2 | 3.4 |
| Marital status | | | | | | |
| Married | 89,461 | 2.6 | 12.8 | 73.7 | 8.3 | 2.6 |
| Unmarried | 3,490 | 4.2 | 16.7 | 68.9 | 7.8 | 2.3 |
| Divorced or bereavement | 834 | 3.5 | 15.4 | 67.0 | 10.3 | 3.8 |
| Missing | 2,966 | 3.4 | 14.8 | 70.0 | 8.3 | 3.5 |
| Occupational status | | | | | | |
| Employed | 61,625 | 2.6 | 12.8 | 74.2 | 8.1 | 2.3 |
| Housewife or unemployed | 29,276 | 2.9 | 13.1 | 72.1 | 9.0 | 3.0 |
| Student | 557 | 3.1 | 16.9 | 74.3 | 5.0 | 0.7 |
| Missing | 5,293 | 3.2 | 14.3 | 70.9 | 8.1 | 3.5 |
| Household income (million yen) | | | | | | |
| ≤199 | 4,851 | 3.3 | 13.7 | 67.1 | 11.3 | 4.7 |
| 200–399 | 29,548 | 3.0 | 13.0 | 71.8 | 9.2 | 3.0 |
| 400–599 | 28,340 | 2.4 | 12.4 | 74.8 | 8.1 | 2.4 |
| 600–799 | 13,713 | 2.2 | 12.5 | 76.5 | 7.0 | 1.8 |
| 800–899 | 5,710 | 2.1 | 13.5 | 76.5 | 6.3 | 1.6 |
| ≥1,000 | 3,676 | 2.6 | 14.2 | 75.7 | 6.2 | 1.3 |
| Missing | 10,913 | 3.7 | 14.2 | 70.4 | 8.7 | 2.9 |
| Educational attainment | | | | | | |
| ECD1 | 4,415 | 4.2 | 13.8 | 65.3 | 11.8 | 5.0 |
| ECD2 | 28,843 | 3.0 | 12.5 | 70.9 | 10.0 | 3.5 |
| ECD3 | 38,706 | 2.4 | 12.6 | 74.6 | 8.1 | 2.2 |
| ECD4 | 20,029 | 2.4 | 14.1 | 76.9 | 5.5 | 1.1 |

*(Continued)*

**Table 1.** (Continued)

| | | Severe-moderate underweight | Mild underweight | Normal | Overweight | Obesity |
|---|---|---|---|---|---|---|
| | | (BMI≤16.9 kg/m²) | (BMI: 17.0–18.4 kg/m²) | (BMI: 18.5–24.9 kg/m²) | (BMI: 25.0–29.9 kg/m²) | (BMI≥30.0 kg/m²) |
| | | (N = 2,640, 2.7%) | (N = 12,564, 13.0%) | (N = 70,995, 73.4%) | (N = 8,057, 8.3%) | (N = 2,495, 2.6%) |
| | N | % | % | % | % | % |
| Missing | 4,758 | 3.3 | 13.7 | 71.0 | 8.5 | 3.5 |

BMI: Body mass index

ECD1: Junior high school; ECD2: High school; ECD3: Technical junior college, technical/vocational college, or associate degree; ECD4: bachelor's degree or postgraduate degree.

**Table 2. Relative ratios of age, parity, smoking, and socioeconomic factors for BMI class (Multiple imputation, N = 96,751).**

| | Severe-moderate underweight | | | Mild underweight | | | Normal | Overweight | | | Obesity | | |
|---|---|---|---|---|---|---|---|---|---|---|---|---|---|
| | aRR | 95%CI | P | aRR | 95%CI | P | Reference | aRR | 95%CI | P | aRR | 95%CI | P |
| Age (years) | | | | | | | | | | | | | |
| <19 | 2.00 | (1.49–2.68) | <0.001 | 1.32 | (1.10–1.59) | 0.003 | - | 0.59 | (0.46–0.77) | <0.001 | 0.19 | (0.10–0.37) | <0.001 |
| 20–24 | 1.54 | (1.35–1.76) | <0.001 | 1.36 | (1.27–1.46) | <0.001 | - | 0.75 | (0.68–0.82) | <0.001 | 0.51 | (0.43–0.61) | <0.001 |
| 25–29 | 1.25 | (1.13–1.38) | <0.001 | 1.16 | (1.10–1.22) | <0.001 | - | 0.85 | (0.80–0.90) | <0.001 | 0.77 | (0.69–0.86) | <0.001 |
| 30–34 | 1.00 | | | 1.00 | | | - | 1.00 | | | 1.00 | | |
| 35–39 | 0.76 | (0.67–0.86) | <0.001 | 0.84 | (0.79–0.89) | <0.001 | - | 1.14 | (1.07–1.21) | <0.001 | 1.17 | (1.05–1.30) | 0.004 |
| >40 | 0.51 | (0.37–0.69) | <0.001 | 0.72 | (0.64–0.82) | <0.001 | - | 1.37 | (1.22–1.54) | <0.001 | 1.26 | (1.03–1.55) | 0.022 |
| Parity | | | | | | | | | | | | | |
| 0 | 1.00 | | | 1.00 | | | - | 1.00 | | | 1.00 | | |
| 1 | 0.88 | (0.80–0.97) | 0.008 | 0.98 | (0.94–1.03) | 0.461 | - | 1.15 | (1.09–1.22) | <0.001 | 1.02 | (0.93–1.13) | 0.629 |
| >2 | 0.84 | (0.74–0.95) | 0.005 | 0.91 | (0.86–0.97) | 0.002 | - | 1.17 | (1.10–1.26) | <0.001 | 1.09 | (0.98–1.22) | 0.120 |
| Smoking | | | | | | | | | | | | | |
| Never-smoking | 1.00 | | | 1.00 | | | - | 1.00 | | | 1.00 | | |
| Quitting smoking before pregnancy | 0.83 | (0.75–0.92) | 0.001 | 0.91 | (0.87–0.96) | <0.001 | - | 1.03 | (0.97–1.09) | 0.295 | 1.19 | (1.08–1.32) | 0.001 |
| Quitting smoking early pregnancy/still smoking | 1.30 | (1.17–1.45) | <0.001 | 1.07 | (1.02–1.14) | 0.011 | - | 1.20 | (1.12–1.28) | <0.001 | 1.41 | (1.27–1.57) | <0.001 |
| Marital status | | | | | | | | | | | | | |
| Married | 1.00 | | | 1.00 | | | - | 1.00 | | | 1.00 | | |
| Unmarried | 1.12 | (0.92–1.35) | 0.251 | 1.17 | (1.06–1.3) | 0.002 | - | 1.00 | (0.88–1.15) | 0.947 | 0.95 | (0.75–1.21) | 0.692 |
| Divorced or bereavement | 1.26 | (0.86–1.86) | 0.237 | 1.34 | (1.1–1.63) | 0.004 | - | 0.83 | (0.66–1.05) | 0.122 | 0.70 | (0.49–1.02) | 0.063 |
| Occupational status | | | | | | | | | | | | | |
| Employed | 1.00 | | | 1.00 | | | - | 1.00 | | | 1.00 | | |
| Housewife or unemployed | 1.26 | (1.15–1.38) | <0.001 | 1.11 | (1.06–1.16) | <0.001 | - | 1.03 | (0.98–1.09) | 0.250 | 1.14 | (1.04–1.25) | 0.004 |

(Continued)

**Table 2.** (Continued)

| | Severe-moderate underweight | | | Mild underweight | | | Normal | Overweight | | | Obesity | | |
|---|---|---|---|---|---|---|---|---|---|---|---|---|---|
| | aRR | 95%CI | P | aRR | 95%CI | P | Reference | aRR | 95%CI | P | aRR | 95%CI | P |
| Student | 0.65 | (0.39–1.08) | 0.098 | 1.00 | (0.79–1.28) | 0.982 | - | 0.74 | (0.50–1.11) | 0.145 | 0.58 | (0.21–1.57) | 0.279 |
| Household income (million yen) | | | | | | | | | | | | | |
| <199 | 0.88 | (0.67–1.14) | 0.326 | 0.91 | (0.80–1.04) | 0.165 | - | 1.64 | (1.39–1.94) | <0.001 | 2.84 | (2.05–3.91) | <0.001 |
| 200–399 | 0.88 | (0.71–1.09) | 0.254 | 0.89 | (0.80–0.98) | 0.020 | - | 1.34 | (1.16–1.55) | <0.001 | 1.89 | (1.40–2.54) | <0.001 |
| 400–599 | 0.80 | (0.64–0.99) | 0.043 | 0.86 | (0.78–0.95) | 0.003 | - | 1.17 | (1.01–1.35) | 0.034 | 1.53 | (1.14–2.07) | 0.005 |
| 600–799 | 0.80 | (0.64–1.01) | 0.056 | 0.87 | (0.78–0.97) | 0.009 | - | 1.07 | (0.92–1.24) | 0.401 | 1.29 | (0.94–1.77) | 0.108 |
| 800–899 | 0.81 | (0.62–1.06) | 0.124 | 0.95 | (0.84–1.06) | 0.348 | - | 0.98 | (0.83–1.17) | 0.864 | 1.22 | (0.86–1.74) | 0.263 |
| >1,000 | 1.00 | | | 1.00 | | | - | 1.00 | | | 1.00 | | |
| Educational attainment | | | | | | | | | | | | | |
| ECD1 | 1.27 | (1.04–1.55) | 0.020 | 0.93 | (0.83–1.03) | 0.172 | - | 2.18 | (1.92–2.47) | <0.001 | 3.97 | (3.22–4.89) | <0.001 |
| ECD2 | 1.11 | (0.98–1.26) | 0.105 | 0.89 | (0.84–0.94) | <0.001 | - | 1.77 | (1.64–1.91) | <0.001 | 2.72 | (2.34–3.16) | <0.001 |
| ECD3 | 0.99 | (0.88–1.11) | 0.823 | 0.91 | (0.87–0.96) | 0.001 | - | 1.42 | (1.32–1.53) | <0.001 | 1.73 | (1.49–2.01) | <0.001 |
| ECD4 | 1.00 | | | 1.00 | | | - | 1.00 | | | 1.00 | | |

BMI: Body mass index

aRR: adjusted relative ratio

CI: confidence interval

ECD1: Junior high school; ECD2: High school; ECD3: Technical junior college, technical/vocational college, or associate degree; ECD4: bachelor's degree or postgraduate degree.

obesity. This was the first study to report the association of socioeconomic factors with not having a normal BMI, including severe to moderate underweight, with age- and parity-adjusted multivariable analysis among pregnant Japanese women.

A review reported that the inverse association between educational attainment and obesity was common in developed countries [11]. Studies of pregnant women in France and women aged 20–64 years in Japan [10, 17] indicated a clear inverse association with the largest point estimate aRR for obesity (3.97) and obesity (2.18). Household income had the second-largest point estimate aRRs for obesity (2.84) and obesity (1.64) in our study and had clear inverse associations similar to those in French and Japanese studies [10, 17]. Low SES often passes through the following generation [33] and is linked to the low health literacy [34]. Therefore, pregnant women with low SES may also have a low SES during childhood, which is related to poor dietary and physical activity habits [17, 35]. Furthermore, after adulthood, healthier diets often cost more, which may result in reduced consumption of vegetables and fruits and increased consumption of ultra-processed diets [36, 37].

However, regarding the adverse effects of low educational attainment and household income on severe-moderate and mildly underweight, only ECD1 had a significantly higher aRR for severe-moderate underweight. Considering the significantly protective aRRs of ECD2 and 3, and 200–399, 400–599, 600–799-million-yen household income for mild underweight, and

400–599-million-yen household income for severe-moderate overweight, the highest educational attainment and household income categories had higher risks. Studies among women aged 20–64 years in Japan and 25–69 years in Korea reported that educational attainment and household income had no statistical significance in underweight women [6, 22]. However, these two studies' sample sizes were smaller than ours' (N = 5,004 and 1,410 vs. 96,751). Thus, a possible reason for the significant RRs is the large sample size, which revealed a small but significant effect. Underweight is prevalent among Japanese women [1–3], and young women have a strong desire for thinness, with an ideal BMI of 18.59 kg/m$^2$ [38]. The lower ideal BMI may surpass the healthy BMI even among pregnant women with the highest educational attainment, who seem to have the highest health literacy. However, the lack of significant RR of ECD4 for severe-to-moderate weight may be because they should avoid extremely low body weight. Meanwhile, the significantly higher RR of ECD1 for severe-to-moderate underweight may be due to lower health literacy rather than lower income, as the RR was adjusted for household income.

Although occupational status had a negative result in recent Japanese and Korean studies among the general population [6, 22], homemakers and unemployed women had significantly higher RRs for obesity, mild, and severe-moderate underweight in our study, which may be due to the larger sample size. Stress caused by unemployment is linked to changes in eating behavior in both directions, including appetite loss and stress eating, and unemployment is associated with a reduction in daily physical activity [39–41]. Unemployment may also result from obesity because obese and overweight individuals, especially younger women, have more difficulty getting a job than non-obese or overweight individuals [42]. These factors are intricately intertwined with BMI, creating a U-shaped relationship between homemakers and the unemployed in the BMI categories.

Unmarried, divorced, or bereaved women had significantly higher RRs for mild underweight but not for severe-moderate underweight, overweight, or obese women. Unmarried pregnancies, especially unintended pregnancies, can lead to emotional, social, or financial difficulties [43], and divorce and bereavement are stressful events [44].

Quitting smoking in early pregnancy/still smoking had significantly higher RRs for all four not having a normal BMI. People believe smoking is an effective way to reduce body weight [45], and girls who are overweight or obese are more likely to initiate smoking [46]. As our analysis was cross-sectional, the higher RRs for overweight and obesity were reflected in overweight or obese women who wanted to lose weight by smoking. Furthermore, because many studies reported that smoker weights were fewer than nonsmokers and former smokers [47], the higher RRs for severe-to-moderate underweight and mild underweight were considered to be reflected by women who were underweight due to the cigarette effect and were unable to quit smoking. Meanwhile, quitting smoking before pregnancy had a higher RR for obese individuals and lower RRs for severe-to-moderate underweight and mildly underweight individuals. These relationships are probably due to the association between smoking cessation and weight gain. Quitting smoking before pregnancy is also a risk factor for obesity.

Age and parity were used as adjusted variables. Older age and parity have been reported to be associated with a higher BMI [48, 49]. Thus, our results for age and parity RRs are compatible with those of previous studies.

A strength of the present study was that BMI was divided into five categories, including severe-moderate underweight, which were more prevalent among East Asians, using large-scale cohort data, and the RRs were obtained in multivariable analysis. However, it should be noted that the findings from Japan, which has a relatively high number of underweight women and is the only high-income country with a high prevalence of underweight individuals, may not be applicable to low-income countries or countries with a low prevalence of underweight women [50].

This study has several limitations. First, as this study obtained cross-sectional results, we needed conclusive evidence of causal relationships. Second, because the participants were pregnant Japanese women whose average BMI was low compared to non-East Asian countries, the generalization of the results of our study would be limited. Third, childhood SES is related to later obesity [51], but our study did not identify these factors. Fourth, heated tobacco products were introduced in Japan in 2014, and the prevalence of heated tobacco product use in Japan increased from 0.2% in 2014 to 11.3% in 2019 [52]. However, this study did not ask about heated tobacco product use because the recruitment period was from 2011 to 2014. Finally, the variables data were obtained from self-administered questionnaires. The validities of the used questions were directly evaluated, but questions about smoking habits and alcohol consumption were based on the question used in the Japan Public Health Centre-based prospective Study for the Next Generation [JPHC-NEXT] [53], and occupation in early pregnancy was based on the 2009 Japan Standard Occupational Classification [28].

## Conclusions

Lower education and household income was associated with being overweight and obese, and those with the lowest education and household income had relatively higher point estimate RRs. Junior high school education was associated with a higher risk of severe to moderate underweight. Homemakers or unemployed were associated with a higher risk of severe-moderate underweight, overweight, and obesity. Being unmarried, divorced, or bereaved were significant risk factors for mild underweight status. Quitting smoking early in pregnancy/still smoking was a risk factor for all four not having normal BMIs, but quitting smoking before pregnancy was the only risk factor for obesity. Lower educational attainment, smoking, and household income are important intervention targets for obesity and severe-moderate underweight prevention in younger women.

## Supporting information

**S1 Table. Relative ratios of age, parity, smoking, and socioeconomic factors by BMI class (complete dataset, N = 78,682).**
(DOCX)

## Acknowledgments

We want to express our gratitude to all the participants of the JECS and all staff members involved in the data collection. We would like to thank Editage (www.editage.com) for the English language editing.

Members of the JECS Group as of 2023: Michihiro Kamijima (principal investigator, Nagoya City University, Nagoya, Japan), Shin Yamazaki (National Institute for Environmental Studies, Tsukuba, Japan), Yukihiro Ohya (National Center for Child Health and Development, Tokyo, Japan), Reiko Kishi (Hokkaido University, Sapporo, Japan), Nobuo Yaegashi (Tohoku University, Sendai, Japan), Koichi Hashimoto (Fukushima Medical University, Fukushima, Japan), Chisato Mori (Chiba University, Chiba, Japan), Shuichi Ito (Yokohama City University, Yokohama, Japan), Zentaro Yamagata (University of Yamanashi, Chuo, Japan), Hidekuni Inadera (University of Toyama, Toyama, Japan), Takeo Nakayama (Kyoto University, Kyoto, Japan), Tomotaka Sobue (Osaka University, Suita, Japan), Masayuki Shima (Hyogo Medical University, Nishinomiya, Japan), Seiji Kageyama (Tottori University, Yonago, Japan), Narufumi Suganuma (Kochi University, Nankoku, Japan), Shoichi Ohga (Kyushu University, Fukuoka, Japan), and Takahiko Katoh (Kumamoto University, Kumamoto, Japan).

## Author Contributions

**Conceptualization:** Yasuaki Saijo, Yuki Kunori.

**Formal analysis:** Yasuaki Saijo.

**Investigation:** Yasuaki Saijo, Eiji Yoshioka, Yoshiya Ito, Reiko Kishi.

**Methodology:** Yasuaki Saijo.

**Writing – original draft:** Yasuaki Saijo.

**Writing – review & editing:** Eiji Yoshioka, Yukihiro Sato, Yuki Kunori, Tomoko Kanaya, Kentaro Nakanishi, Yasuhito Kato, Ken Nagaya, Satoru Takahashi, Yoshiya Ito, Hiroyoshi Iwata, Takeshi Yamaguchi, Chihiro Miyashita, Sachiko Itoh, Reiko Kishi.

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
