## [Decision Letter · Decision Letter 0]

10 Apr 2024

PONE-D-24-01959Maternal pre-pregnancy body mass index and related factors: a cross-sectional analysis from the Japan Environment and Children’s StudyPLOS ONE

Dear Dr. Saijo

Thank you for submitting your manuscript to PLOS ONE. After careful consideration, we feel that it has merit but does not fully meet PLOS ONE’s publication criteria as it currently stands. Therefore, we invite you to submit a revised version of the manuscript that addresses the points raised during the review process.

We look forward to receiving your revised manuscript.

Kind regards,

Malshani Lakshika Pathirathna, PhD

Academic Editor

PLOS ONE

Journal Requirements:

"This study was funded by the Ministry of Environment, Japan. The findings and conclusions of this article are solely the responsibility of the authors and do not represent the government's official views."

Reviewers' comments:

Reviewer's Responses to Questions

**Comments to the Author**

1. Is the manuscript technically sound, and do the data support the conclusions?

Reviewer #1: Yes

Reviewer #2: Partly

2. Has the statistical analysis been performed appropriately and rigorously? 

Reviewer #1: Yes

Reviewer #2: Yes

3. Have the authors made all data underlying the findings in their manuscript fully available?

Reviewer #1: No

Reviewer #2: Yes

4. Is the manuscript presented in an intelligible fashion and written in standard English?

Reviewer #1: Yes

Reviewer #2: Yes

5. Review Comments to the Author

Reviewer #1: There are some spelling and grammatical errors in the document, please review:

e.g. Serioeconomic status page 5, line 73

I am also uncomfortable with the term 'inappropriate' for anyone who does not have a normal BMI.

Perhaps a better way to state this would be those without a normal BMI. This is in line with Equality, Diversity and Inclusivity terms of peer-reviewed research in current times.

It is very interesting to see such a dichotomy in results from the Japanese population with such a high number of 'normal BMI' participants. I think this is worth highlighting in the discussion with some references on global values. While it cannot be generalized, it is a result that is uncommon.

I felt that the population of still smoking and quitting at early pregnancy should be divided as the biochemical differences in these women are so different. It would add value to the paper. It is also important to talk about the J shaped RRs for some exposures such as smoking and BMI.

Otherwise a well written paper with minor errors.

Reviewer #2: Dear Editor,

Thanks for inviting me to review this study. The study was interesting to read and had some relevant information that will add to the existing literature. However, it is not suitable for publication in its current form. Please find my comments below.

Introduction

There was no clear justification for the need for the study.

Method

The study lacked clear step by step procedure used for data collection.

It is not clear how the hospitals included in the study were selected.

It is not clear if consent was obtained, or the type of consent obtained prior to data collection.

There was no clear information about inclusion and exclusion criteria.

How were the pregnant women recruited?

It is not clear if the study was done retrospectively.

There was no clear information about the questionnaire used in the study, how was developed? was it validated and how it was validated.

What confounding factors were detected and how were they managed?

It is not clear how pre pregnancy BMI of the pregnant women were obtained since they were already pregnant and only pregnant women were included in the study?

Was it a self-reported BMI or were participants’ BMI taken before they got pregnant?

The authors failed to provide information on how the parameter including BMI, Height and weight were obtained from the participants.

Including duration of smoking could have benefitted the study.

Result

The tables are two long could have been better presented with graphs.

Recommendation

Overall, the study has large sample size and some novel information that could add to literature, but it is not suitable for publication in its current form. There is a need to review the methodology.

6. PLOS authors have the option to publish the peer review history of their article (what does this mean?). If published, this will include your full peer review and any attached files.

Reviewer #1: **Yes: **Nasloon Ali

Reviewer #2: **Yes: **Ngozika Ezinne

---

## [Author Response · Author response to Decision Letter 0]

30 Apr 2024

#Answer to Journal Requirements:

[Response]

We revised the manuscript to comply with PLOS ONE's file naming and style requirements.

"This study was funded by the Ministry of Environment, Japan. The findings and conclusions of this article are solely the responsibility of the authors and do not represent the government's official views."

[Response]

The statement was changed as follow, and it was included in the cover letter.

“This study was funded by the Ministry of Environment, Japan. The findings and conclusions of this article are solely the responsibility of the authors and do not represent the government's official views. The funders had no role in study design, data collection and analysis, decision to publish, or preparation of the manuscript.”

[Response]

The ethics statement was mentioned only in the Method section, and not in any other part of the document.

#Answer to Reviewers' comments:

Reviewer #1: There are some spelling and grammatical errors in the document, please review:

e.g. Serioeconomic status page 5, line 73

I am also uncomfortable with the term 'inappropriate' for anyone who does not have a normal BMI.

Perhaps a better way to state this would be those without a normal BMI. This is in line with Equality, Diversity and Inclusivity terms of peer-reviewed research in current times.

[Response]

I have made sure to check all the spellings thoroughly. Additionally, I have replaced every instance of the word 'inappropriate' with 'not having normal', and made other necessary changes wherever applicable.

It is very interesting to see such a dichotomy in results from the Japanese population with such a high number of 'normal BMI' participants. I think this is worth highlighting in the discussion with some references on global values. While it cannot be generalized, it is a result that is uncommon.

[Response]

P21L286: “However, it should be noted that the findings from Japan, which has a relatively high number of underweight women and is the only high-income country with a high prevalence of underweight individuals, may not be applicable to low-income countries or countries with a low prevalence of underweight women [50].” was added.

A new reference was added as the 50th reference: NCD Risk Factor Collaboration (NCD-RisC). Worldwide trends in underweight and obesity from 1990 to 2022: a pooled analysis of 3663 population-representative studies with 222 million children, adolescents, and adults. Lancet. 2024 Mar 16;403(10431):1027-1050. doi: 10.1016/S0140-6736(23)02750-2. Epub 2024 Feb 29. 

I felt that the population of still smoking and quitting at early pregnancy should be divided as the biochemical differences in these women are so different. It would add value to the paper. It is also important to talk about the J shaped RRs for some exposures such as smoking and BMI.

[Response]

Since smoking cessation after pregnancy occurred after the outcome measure for pre-pregnancy BMI, we considered factors that occurred after the outcome measure should not be treated as causal factors.

Otherwise a well written paper with minor errors.

[Response]

Thank you for your valuable and constructive comments.

Reviewer #2: Dear Editor,

Thanks for inviting me to review this study. The study was interesting to read and had some relevant information that will add to the existing literature. However, it is not suitable for publication in its current form. Please find my comments below.

Introduction

There was no clear justification for the need for the study.

[Response]

P4L85: “Obesity and being underweight during pregnancy increase the risk of adverse obstetric outcomes. Identifying related factors can help prevent these outcomes.” was added.

Method

The study lacked clear step by step procedure used for data collection.

[Response]

Details of the data collection procedure were added. Please see the following responses.

It is not clear how the hospitals included in the study were selected.

[Response]

P6L115: “Recruitment activities were conducted at healthcare providers and local government facilities to identify eligible women in the study areas. However, the recruitment was not entirely random. Our team made every effort to reach out to as many eligible women in the study areas as possible. The child coverage was estimated to be approximately 45% of the studied areas [28, 29].” was added.

It is not clear if consent was obtained, or the type of consent obtained prior to data collection.

[Response]

P5L104: “and written informed consent was obtained from all participants prior to data collection.” was added.

There was no clear information about inclusion and exclusion criteria.

[Response]

P4L104: “Written informed consent was obtained from all participants prior to data collection. The inclusion criteria were as follows: 1) Pregnant women whose expected delivery dates were between August 2011 and March 2014, 2) pregnant women who resided in one of the study areas selected by the Regional Centres at the time of recruitment, and who were expected to reside continually in Japan for the foreseeable future, and 3) pregnant women who visited a cooperating health care provider selected by the Regional Centre or local government offices to obtain a Mother-Child Health Handbook in a study area during the recruiting period. The exclusion criteria were as follows: 1) Pregnant women who did not consent to participate in the study, 2) pregnant women who showed difficulty in comprehending the study procedures or filling out the questionnaires without support, and 3) pregnant women who were reportedly not accessible at the time of delivery (e.g., women who planned to give birth outside the study area).” was added.

How were the pregnant women recruited?

[Response]

Please see the above responses.

It is not clear if the study was done retrospectively.

[Response]

P5L101: “The protocol for analyzing the data in this study was prepared after the relevant data was collected.” was added.

There was no clear information about the questionnaire used in the study, how was developed? was it validated and how it was validated.

[Response]

P20L297 “Finally, the variables data were obtained from self-administered questionnaires. The validities of the used questions were directly evaluated, but questions about smoking habits and alcohol consumption were based on the question used in the Japan Public Health Centre-based prospective Study for the Next Generation [JPHC-NEXT] [53], and occupation in early pregnancy was based on the 2009 Japan Standard Occupational Classification [28].” was added.

A new reference was added: Yokoyama Y, Takachi R, Ishihara J, Ishii Y, Sasazuki S, Sawada N, Shinozawa Y, Tanaka J, Kato E, Kitamura K, Nakamura K, Tsugane S. Validity of Short and Long Self-Administered Food Frequency Questionnaires in Ranking Dietary Intake in Middle-Aged and Elderly Japanese in the Japan Public Health Center-Based Prospective Study for the Next Generation (JPHC-NEXT) Protocol Area. J Epidemiol. 2016 Aug 5;26(8):420-32. doi: 10.2188/jea.JE20150064. 

What confounding factors were detected and how were they managed? 

[Response]

P7L157: “Maternal age and parity were selected as covariates, and data were transcribed from the medical records. Maternal age was categorized as <19, 20–24, 25–29, 30–34, 35–39, or >40 years, and parity was ranked as zero, one, two, or more.”

was changed to

“Maternal age and parity were selected as covariates (confounders) based on previous research on pregnant women [17, 32], and data were transcribed from the medical records. Maternal age was categorized as <19, 20–24, 25–29, 30–34, 35–39, or >40 years, and parity was ranked as zero, one, two, or more.”

P8L169: “The models included educational attainment, household income, marital status, occupational status, smoking status, age, and parity.”

was changed to

“The models included educational attainment, household income, marital status, occupational status, smoking status, and covariates (age and parity).”

It is not clear how pre pregnancy BMI of the pregnant women were obtained since they were already pregnant and only pregnant women were included in the study?

Was it a self-reported BMI or were participants’ BMI taken before they got pregnant?

The authors failed to provide information on how the parameter including BMI, Height and weight were obtained from the participants.

[Response]

P6L133: “Maternal height and pre-pregnancy weight were obtained from medical records. If missing, these data were obtained from self-reports.” was added.

Including duration of smoking could have benefitted the study.

[Response]

Because this paper dealt with many variables, we considered that further categories of smoking would lead to an incomprehensible table, and the number of each category would be rather small. So, we did not conduct a further analysis of the smoking duration.

Result

The tables are two long could have been better presented with graphs.

[Response]

We would like to present precise 95% confidence intervals. Further, to compare with two previous studies about pregnant women and BMI which had a table of main results, we considered that the current table was more appropriate for this paper.

Recommendation

Overall, the study has large sample size and some novel information that could add to literature, but it is not suitable for publication in its current form. There is a need to review the methodology.

[Response]

Thank you for your valuable and constructive comments. We amended the method section according to the reviewers’ comments as above.

---

## [Decision Letter · Decision Letter 1]

21 May 2024

Maternal pre-pregnancy body mass index and related factors: a cross-sectional analysis from the Japan Environment and Children’s Study

PONE-D-24-01959R1

Dear Prof. Yasuaki Saijo,

We’re pleased to inform you that your manuscript has been judged scientifically suitable for publication and will be formally accepted for publication once it meets all outstanding technical requirements.

Kind regards,

Malshani Lakshika Pathirathna, PhD

Academic Editor

PLOS ONE

Additional Editor Comments (optional):

Reviewers' comments:

Reviewer's Responses to Questions

**Comments to the Author**

1. If the authors have adequately addressed your comments raised in a previous round of review and you feel that this manuscript is now acceptable for publication, you may indicate that here to bypass the “Comments to the Author” section, enter your conflict of interest statement in the “Confidential to Editor” section, and submit your "Accept" recommendation.

Reviewer #1: All comments have been addressed

Reviewer #2: All comments have been addressed

2. Is the manuscript technically sound, and do the data support the conclusions?

Reviewer #1: Yes

Reviewer #2: Yes

3. Has the statistical analysis been performed appropriately and rigorously? 

Reviewer #1: Yes

Reviewer #2: Yes

4. Have the authors made all data underlying the findings in their manuscript fully available?

Reviewer #1: Yes

Reviewer #2: Yes

5. Is the manuscript presented in an intelligible fashion and written in standard English?

Reviewer #1: Yes

Reviewer #2: Yes

6. Review Comments to the Author

Reviewer #1: Thank you for your responses. I believe you have answered my questions appropriately. I find the article sound for publication.

Reviewer #2: (No Response)

7. PLOS authors have the option to publish the peer review history of their article (what does this mean?). If published, this will include your full peer review and any attached files.

Reviewer #1: No

Reviewer #2: **Yes: **Dr Ngozika Esther Ezinne

---

## [Editor Report · Acceptance letter]

24 May 2024

PONE-D-24-01959R1 

PLOS ONE

Dear Dr. Saijo, 

I'm pleased to inform you that your manuscript has been deemed suitable for publication in PLOS ONE. Congratulations! Your manuscript is now being handed over to our production team.

Kind regards, 

on behalf of

Dr. Malshani Lakshika Pathirathna 

Academic Editor

PLOS ONE